# Validation, Reliability, and Usefulness of the Functional Agility Square Test [FAST]

**DOI:** 10.3390/jfmk10020126

**Published:** 2025-04-10

**Authors:** Romina Müller, Daniel Büchel, Jochen Baumeister

**Affiliations:** Exercise Science and Neuroscience Unit, Department of Exercise and Health, Faculty of Science, Paderborn University, 33098 Paderborn, Germany; romina.mueller@uni-paderborn.de (R.M.); daniel.buechel@uni-paderborn.de (D.B.)

**Keywords:** testing, cognition, team sports, agility

## Abstract

**Background**: Agility is crucial in game sports, requiring both motor and cognitive skills. Athletes must perceive and process information to adapt movements, yet traditional agility tests often lack cognitive and multidirectional demands. Additionally, modern test systems are mostly stationary. This study evaluated the novel and portable “Functional Agility Square Test” (FAST) for validity, reliability, and usefulness. **Methods**: To assess discriminant validity, 22 game sports (GS) and 22 non-game sports (NGS) athletes participated in one session. Test–retest reliability was examined with 36 GS athletes (20 female) across three sessions. Participants performed cognitive (FAST_COG), preplanned (FAST_MOT), and randomized (FAST_SAT) reactive change-of-direction tasks, each repeated three times per session. **Results**: Results showed significantly lower response times (RTs) in GS compared to NGS (*p* < 0.05). Mean RTs indicated moderate relative reliability (ICC 0.50–0.74), while medians showed moderate to good reliability (ICC 0.59–0.83). Usefulness was evident from the first session (FAST_MOT) or from the third session (FAST_SAT) based on median RTs. **Conclusions**: Thus, the FAST seems to be valid, reliable, and sensitive for GS-based agility assessment. Its portable setup enables ecologically valid field testing. Future research should further increase task complexity to better simulate game conditions.

## 1. Introduction

Game sports take place in enriched and dynamic environments. Athletes need to screen and adapt to these sporting surroundings by perceiving and processing sensory cues such as the routes of teammates and opponents or instructions from their coaches/teammates. Accordingly, the ability to perform whole-body movements with a change in velocity or direction in response to a stimulus [1]—also referred to as agility—is a key skill in team sports. Agility performance is influenced by motor-related aspects such as linear sprint speed, leg muscle quality, core strength, and cognitive-related aspects, like visual scanning, knowledge of the situations, pattern recognition, and anticipation [1,2,3,4,5].

Most commonly, test setups for change in direction (COD) or agility testing in team sports use a Y-shape, including one preplanned or reactive reaction at a 45° angle. However, game sport athletes must visually scan their environment, process information from all directions, and adapt their movements to respond effectively to situational demands [6,7]. This requires executing reactive changes in direction at various angles, including lateral, diagonal, and backward movements, while adjusting running patterns and body positions to meet the complex requirements of the game [8,9,10]. In addition, decision-making behavior in games is much more complex than a reaction that merely involves a decision to initiate a change in direction to the left or the right [11]. Furthermore, major limitations of existing agility tests are the use of the total time as an outcome parameter, which favors linear sprint ability and, thus, underestimates agility performance, the neglect of important contextual factors such as entry speed, change in direction angle, and intention to move, and the lack of comprehensive validity across different athlete populations [12].

More recent agility tests increase the possible degrees of directional changes and enable agility tests with multidirectional movements [13,14,15,16,17,18]. Furthermore, tests such as the “Star Run” offer cognitively demanding test settings with customized color-coded signals to instruct different movement tasks [14] or dual-task exercises in which cognitive and motor tasks are combined [17]. Despite increased motor (various running directions) and cognitive (higher order cognitive demands) degrees of freedom, these tests demonstrated good to excellent reliability [14,15,16]. However, such tests frequently take place on stationary systems, and lab-based systems, which lack portability and cannot be used simultaneously in varying sports facilities, such as the soccer court or sports hall. Mobile light-based training and testing systems are emerging, allowing for more variable and ecologically valid test setups. The portability of the sensors allows for a sports-specific alignment in the test setups, related to angles, distances, and surfaces [19,20,21,22,23]. Furthermore, light-based agility systems have been reported as reliable and discriminant valid measurement systems for agility testing [24,25,26]. Therefore, light-based tests seem appropriate portable alternatives for the aforementioned stationary testing methods, but validated tests using assessable light-based training systems are sparse.

To overcome these shortcomings of limited multidirectionality and simplicity, we developed the “Functional Agility Square Test” (FAST). This light-based agility test aims to test the reactive agility of team sports athletes in a mobile and ecologically valid manner. The primary aim of the present investigation was to examine the discriminant validity, reliability (reproducibility), and usefulness (sensitivity to identify changes in performance) of the FAST in samples of healthy athletes. A group comparison between game and non-game sports athletes was performed to measure discriminant validity. This subtype of validity refers to the extent to which a test or a measure accurately distinguishes between constructs or variables that are theoretically expected to be distinct [24]. Since game sports athletes are habituated to reactive multidirectional runs, we hypothesized that game sports athletes outperform non-game sports athletes in the FAST. To analyze the FAST’s test–retest reliability and usefulness, a group of healthy game sports athletes performed the FAST three times with at least a one-week break between the sessions. The test–retest reliability describes the reproducibility of test scores from the same person in two or more test sessions [27]. Furthermore, usefulness describes the sensitivity of a test to detect performance changes [15,16]. As response times are often skewed [28], the measure of central tendency should be considered for reporting response times. Therefore, the secondary aim of this study was to investigate possible differences in mean and median response times as standard measures of central tendency.

## 2. Materials and Methods

### 2.1. Experimental Approach

This study combined a between-subject and within-subject experimental design for assessing the discriminant validity, test–retest reliability, and usefulness of the FAST. In the between-subject design, two groups of female game sports (GS) and non-game sports athletes (NGS) were tested once to determine discriminant validity. To assess test–retest reliability, a sample of male and female GS was tested at three time points with at least one week of break in between.

### 2.2. Functional Agility Square Test (FAST)

All testing was performed indoors in a lab of the affiliated university. All FAST conditions were executed with the Fitlight System (FITLIGHT Sports Corp., Aurora, ON, Canada) using the application for mobile devices (App Version 2.5.7a) with the Wifi Light System (Version 1.31). The Fitlight System has shown high reliability in COD and reactive agility testing [25,29]. The test setup (Figure 1A) consists of a small square (0.60 m × 0.60 m), which was used as the setup for the FAST_cognitive (FAST_COG) and as the starting position for the following conditions. Furthermore, a larger square (3.5 m × 3.5 m) was used for FAST_motor (FAST_MOT) and the “Simple Agility Test” (FAST_SAT). The Fitlights were positioned in the corners of the square [14,16]. The sensors were placed upright in approximately 6 cm high cones in the latter setup. In addition to these lights, synchronized lights were placed in each corner 70 cm apart and with the LED facing the center position, to ensure that the participants were able to recognize the lights correctly (Figure 1B). In every condition, one corner of the square lit up green and beeped. The remaining Fitlights lit up red and did not present sound. The Fitlights were deactivated through a step near the Fitlight, which activated the proximity sensor (up to 80 cm). Participants were instructed to deactivate the targeted light as fast as possible. In each condition, a sequence of eight lights (each corner two times) had to be deactivated per trial. Between each stimulus, a delay of approximately two seconds was set to provide sufficient time to return to the center of the square.

The FAST is structured as a test battery, assessing the two main components (cognitive- and motor-related aspects) of agility according to [1] in two isolated test conditions (FAST_COG and FAST_MOT), but also as a complex condition requiring the integration of both motor and cognitive components (FAST_SAT) [30]. Therefore, the first condition was the FAST_COG, aiming for the assessment of cognitive response times (RT) with limited motor demands. In this condition, four Fitlights were placed on the floor on the corners of the small square. The participants stood in the middle of the square and had to eliminate the appearing and beeping lights through a swipe movement with their left or right foot as fast as possible. Previous studies have shown that the FITLIGHTS allow for a valid assessment of lower extremity reaction times [31]. The starting position for the further conditions was the aforementioned smaller square placed in the middle of the larger square. The FAST_MOT was used for the assessment of preplanned agility as a task with high-motor, but limited cognitive demands. Therefore, the athletes were aware of the light order (clockwise starting with the frontal left light), and each light was illuminated twice. In contrast, the FAST_SAT aimed to assess reactive agility as a task with high motor and high cognitive demands. During the FAST_SAT, the order of lights lighting up was randomized, using the same number of activated lights (in total, eight lights for each run and each corner twice). Participants were instructed to deactivate the target light with a step next to the light as fast as possible and then return to the starting position as quickly as possible and wait for the next stimulus for the FAST_MOT and FAST_SAT.

A warm-up (approximately 10 min), consisting of self-paced jogging or stationary bike-riding and individually needed additional exercises, was carried out before the measurements. Each test condition of the FAST was performed three times (Figure 2). There was a one-minute break between the trials of the FAST_COG and a two-minute break between each of the trials of the FAST_MOT and FAST_SAT, respectively. The order of the test conditions was randomized for each measurement. To quantify FAST performance, the mean and median RT (time between activation and deactivation of the target light) in milliseconds was extracted for each trial. Trials with measurement errors (e.g., response times < 100 ms or not correctly deactivated stimuli) were deleted before further data analysis. Following Sekulic et al. (2014), a “reactive agility index” (RAI) was calculated as the difference between the best FAST_MOT and the best FAST_SAT performance [32]. For the validity and reliability study, the best of the three trials of each condition was included for statistical analysis.

### 2.3. Participants

For validation purposes, the appropriate sample size was estimated with G*Power, Version 3.1.9.7 [33], to achieve a statistical power of 0.9 at an alpha significance level of 0.05. We based the assumed effect size on Sekulic et al. (2014), who reported significant differences between agility-saturated and non-agility-saturated female athletes in a shorten agility task [32]. The power calculation revealed a required sample size of 19 participants for each group.

In total, 22 female GS and 22 female NGS athletes were included. Athletes were classified as GS if they participated in sports characterized by constantly changing environmental conditions due to interactions with the game equipment (e.g., ball), teammates, and opponents, requiring frequent reactive changes in direction. In contrast, NGS athletes were those engaged in sports where reactive changes in direction were not a primary determinant of performance. All participants performed their sports at a recreational level of around six hours per week. The demographic and anthropometric specifications of the sample are displayed in Table 1.

The guidelines supplied by Bujang and Baharum (2017) were utilized to estimate the necessary sample size to compute Intraclass-Correlation-Coefficients (ICC) for test–retest reliability [34]. A minimum estimate of 0.5 was predicted to indicate at least considerable reliability [35]. The number of participants needed was 11, based on statistical tables with an alpha level of 0.05 and a statistical power of 0.8, assuming the null hypothesis of no agreement between evaluations [34]. In total, the study involved 36 (20 female) healthy GS athletes. All athletes participated in game sports at a regional level and performed their given sports for ~14 years. The demographic and anthropometric specifications of the participants are displayed in Table 1.

Participants were informed about the study and provided their written informed consent. In the case of underage participants (younger than 18 years), a written declaration of consent was obtained from their legal representatives. Participants were excluded from the study if they no longer actively practiced their sport or had recent injuries to the lower extremities. The associated university’s ethics committee approved the study, and the investigation was carried out following the Declaration of Helsinki.

### 2.4. Testing

For evaluating the discriminant validity, the GS and NGS athletes performed three trials of the FAST_COG, FAST_MOT, and FAST_SAT in one session. Testing was carried out as described above (for schematic overview, see Figure 2). To evaluate the test–retest reliability, participants performed the measurements at three time points with at least a one-week break between the sessions (Figure 2). All tests for each participant were carried out at the same time of day to prevent errors resulting from circadian rhythm. The FAST_COG was not included in the test–retest reliability analysis, as this test condition was a scientific approach to categorize the subcomponents of agility, and less meaningful for further practical applied sports investigation.

### 2.5. Statistical Analysis

All statistical tests for discriminant validity analysis were performed with SPSS (version 29). First, the Shapiro–Wilks test was used to test all variables for normal distribution. All variables, except the mean values of the FAST_COG, were normally distributed. For validity examination, differences between GS and NGS were analyzed with the independent *t*-test for normally distributed data and for the mean values of FAST_COG with the Mann–Whitney U test. Additionally, the effect sizes (ES) were assessed, using Cohen’s *d* for normally distributed data and *r* for the FAST_COG mean values. Cohen’s *d* scores were classified as follows: small (d = 0.2), medium (d = 0.50), and large (d = 0.80) and *r* scores were classified as follows: small (r = 0.10), medium (r = 0.30), and large (r = 0.50) [36]. The alpha level for significance was set at *p* < 0.05 for all statistical tests.

Customized scripts for MATLAB 2023a (The MathWorks, Inc., Natick, MA, USA) were used to assess test–retest reliability between the measurements with Intraclass-Correlation-Coefficients (ICC) calculations based on a two-way mixed effects model, single ratings, and absolute agreement. Additionally, the coefficient of variation (CoV) was calculated. ICC scores were classified as follows: poor (<0.50), moderate (0.50–0.75), good (0.75–0.90), and excellent (≥0.90) reliability [35], and CoV < 5% were classified as acceptable [37,38].

For assessing the usefulness (sensitivity to identify training-induced changes in performance) of the test battery, the typical error (TE) and smallest worthwhile change (SWC) were calculated [39]. Tests are categorized as “good” if the SWC value is greater than the TE value, “OK” if the SWC and TE are equal, and “marginal” if the TE is greater than the SWC value [15,39]. The SWC was calculated by multiplying the between-subject standard deviation by 0.5 (SWC 0.5) for testing for performance changes with a medium effect [16,39]. The TE was calculated by dividing the standard deviation of the change scores of the three trials per condition, per day by √2 [16,40].

## 3. Results

### 3.1. Discriminant Validity

Table 2 presents the mean and median RTs for all test conditions (FAST_COG, FAST_MOT, FAST_SAT) and groups (GS and NGS). For both mean and median RTs, significant differences between the two groups in all three FAST conditions have been observed. Statistical analysis revealed significant differences in the FAST_COG (MEAN U = 145, Z = −2.28, *p* = 0.023, ES (r) = −0.34, medium; MEDIAN t(33.89) = −2.47, *p* = 0.009, ES (d) = −0.74, medium), FAST_MOT (MEAN t(42) = −2.32, *p* = 0.013, ES (d) = −0.7, medium; MEDIAN t(42) = −2.17, *p* = 0.018, ES (d) = −0.65, medium), and FAST_SAT (MEAN t(42) = −2.40, *p* = 0.010, ES (d) = −0.73 medium; MEDIAN t(34.93) = −2.06, *p* = 0.023, ES (d) = −0.62, medium). Accordingly, GS demonstrated faster RTs when compared to NGS in all conditions of the FAST. Conversely, the RAI showed no significant differences between the groups, regardless of whether comparing the mean or median values. There, the absolute difference between the FAST_SAT and FAST_MOT was similar across both groups.

### 3.2. Test–Retest Reliability

Overall, the specific conditions of the FAST showed moderate to good reliability, with better reliability scores observed for the FAST_MOT and comparison of median values, as well as the comparison of the second and third session. The complete results of the test–retest reliability are displayed in Table 3 for the mean RTs, and in Table 4 for median RTs.

For the analysis of reliability of mean values, the FAST_MOT and FAST_SAT showed moderate (ICC 0.50–0.74) reliability across the three sessions. The CoV remained below 5% for the FAST_SAT between session one and two (CoV 4.89%) but exceeded the 5% for the FAST_SAT (CoV 5.55%) session two and three and FAST_MOT across all three sessions (CoV 5.68–7.28%). The RAI showed poor to moderate (ICC 0.44–0.56) reliability. CoVs of RAI exceeded 5% (CoV 25.17–25.37%) across the three measurements. For median values, the FAST_MOT and FAST_SAT showed moderate to good (ICC 0.59–0.83) reliability across the three sessions. The CoV remained below 5% for the FAST_MOT and FAST_SAT from session two onwards (CoV 4.48–4.54%). RAI showed poor to moderate (ICC 0.44–0.57) reliability across the three sessions. CoV values of the RAI exceeded 5% (CoV 23.87–23.98%) across all sessions.

### 3.3. Usefulness

Overall, the analysis indicated that the FAST_MOT demonstrated usefulness across all sessions when considering median values and, in session three for mean values, whereas the FAST_SAT showed usefulness only in session three based on median values. The results of the usefulness analysis are displayed in Table 5. As for the reliability analysis, usefulness was analyzed with regard to both the mean and median values of the FAST RTs. For the analysis of mean values of the FAST_MOT, the TE exceeds the SWC 0.5 in session one and session two. In session three, the TE is lower than the SWC 0.5 and, therefore, usefulness can be assumed. For the FAST_SAT, the TE exceeds the SWC 0.5 in all three sessions and, therefore, usefulness cannot be assumed. For the analysis of median values, the TE of the FAST_MOT is lower than the SWC 0.5 in all three sessions and therefore, usefulness can be assumed. For the FAST_SAT, the TE exceeds the SWC 0.5 in session one and two. In session three, the TE is smaller than the SWC 0.5 and, therefore, usefulness can be assumed from session three onwards.

## 4. Discussion

This study investigated the discriminant validity, test–retest reliability, and usefulness of the newly developed FAST. Three main findings emerged. First, significant differences were found between GS and NGS athletes in all three test conditions investigated with the FAST with GS outperforming NGS (lower RTs of GS), independent of using means or medians. Second, differences between the usage of mean and median values were observed. While the mean values demonstrate moderate relative and limited absolute reliability, the FAST conditions revealed moderate to good and acceptable absolute reliability when extracting medians. Third, the usefulness of the FAST can be assumed for both tests from the beginning (FAST_MOT) or at the latest from the third session on (FAST_SAT) when considering median RTs. Therefore, we assume that the FAST is a valid and reliable agility test for team sports, which overcomes current shortcomings of stationarity, lab-based tests and their ecological validity.

### 4.1. Discriminant Validity

As hypothesized, GS showed significantly faster RTs than NGS in the FAST pre-planned and reactive tasks. Therefore, it can be stated that the FAST is a valid test for the assessment of agility in team sports, as faster RTs are related to better agility performance. Previous studies showed similar results for agility tests [22,32]. Our findings align with the study of Sekulic and colleagues (2014) who showed that their shorter version of their agility test was able to distinguish between female athletes of agility-saturated and female athletes of non-agility-saturated sports [32]. Even though the FAST has longer partial and total distances, significant differences were found between female athletes in different sports categories. As we included only female athletes for validation purposes, our chosen running distances could have impacted the significant differences between the two groups. Additionally, Mackala and colleagues (2020) compared individual and team sports athletes in preplanned CODs and reactive agility across four spatial configurations, finding that team sport athletes performed significantly better in three tested conditions. One of these setups (universal reactive agility) is comparable to the FAST and showed significantly better performance of team sports athletes [22]. In terms of athletic performance, it can be stated that training and competition in ever-changing environments for team and game athletes seem to improve agility performance and can, therefore, react faster to signals than individual or non-game sports athletes who are not exposed to these training stimuli. To ensure that the sample was as homogeneous as possible for validation purposes, only female athletes were included to overcome the already proven influence of gender on performance in reactive agility tests [32,41,42,43].

While previous studies reported significant differences between GS and NGS in reactive tasks only [32], we also observed group differences in the pre-planned COD runs. Hence, it might be speculated that the spatial configuration of the FAST probably depicts situations more similar to game sports due to the orientation of the stimuli and running directions within and outside the field of vision. Since NGS are less dependent on active scanning of their sporting environment [22] and the FAST_MOT also includes a reactive component, the experiences in the ever-changing environment may have influenced the behavior of the GS [44]. Based on the current findings, the FAST seems to picture the team sports requirements in female athletes.

### 4.2. Reliability

Overall, the FAST showed moderate to good reliability in the FAST_MOT and FAST_SAT over three test sessions. For the analysis of relative and absolute reliability, the FAST showed divergent results when looking at the mean and median values. Mean RT values revealed only moderate reliability. When looking at the medians, moderate to good reliability was observed. For performance diagnostics, it can be concluded that an extensive familiarization session should be carried out with the athletes so that reliable results can be obtained for the tests regardless of choosing mean or median RTs. In contrast to previously reported reliability studies on agility tests, the reliability of the FAST conditions was relatively low. This might be related to the running courses of the tests, which are more complex than most previously investigated agility tests, often performed in the Y-shape [24]. In the FAST, each trial is composed of eight runs towards the outer lights and eight runs back towards the middle position. Accordingly, each of the turns could potentially lead to measurement errors and, therefore, influence the RT results. These could be explained by, for example, stepping towards the wrong side of the lights and, therefore, not deactivating the sensor at the first attempt, chasing towards the false light (in FAST_SAT), or slight slipping during the cutting movements [32]. These errors in the measurement can affect the reliability and can also explain the differences in the repeatability of the two test conditions, since the more complex test (FAST_SAT) has more scope for movement than the simpler test (FAST_MOT) and, therefore, shows lower ICC values [20,45,46]. Therefore, the test complexity could have impacted the reliability of mean and median RTs.

Studies incorporating comparable agility tasks showed higher ICC values for reactive agility testing like Smith et al. (2024) (time to target, ICC = 0.935). Smith et al. (2024) used the same test equipment (Fitlight System) and a square setup with the starting position in the center of the square [25]. However, the participants ran further distances per run, as (a) the square had a diagonal of 7 m (vs. 5 m in the FAST) and (b) an average of 15 lights (vs. 8 in the FAST) were approached. Thus, Smith et al. (2024) were concerned with long-lasting exhaustive agility runs (60 s) compared to shorter ones (on average 26 s FAST_MOT; 29 s FAST_SAT) in the FAST and, therefore, different load profiles between the studies [25]. Furthermore, a different form of deactivation of the sensors was chosen, since in previous studies, participants were asked to deactivate the lights with their hands [10,20,21,22,23,29]. For the FAST, we asked participants to deactivate the lights with their feet placed next to the Fitlight, since cutting movements are part of the standard game load without reaching with the hand to the ground [30,47]. It remains speculative as to whether the reliability values were impacted by the chosen type of deactivation because, as far as the authors are aware, the selected deactivation method (stepping/cutting movement near to the Fitlight) was used for the first time in a light-based agility configuration.

Another aspect that interacts with the reliability of a test is the degree of ecological validity [48]. To give the athletes no restrictions in the execution of their movements, the participants were given the option of choosing running forwards, backwards, or sideways to the respective target lights [10]. This approach led to the observation that some participants changed their running strategies within a session or between sessions, which could potentially affect day-to-day reliability. For example, some players partially rotated during conditions, so the preorientation varied before starting a run. These changes in running strategies were observed in some athletes; however, other athletes have not changed their running strategies, and the orientation always remained the same. In other established agility tests, which show good to excellent reliability, the running directions are predetermined or there is only one [24]. From this, it can be concluded that running behavior should be examined more closely in future studies, or the body orientation should be specified. These instructions could ensure a better comparability of the results but would limit the ecological validity.

### 4.3. Usefulness

Regarding usefulness, for the mean values, usefulness could only be attested for the FAST_MOT from the third session onwards. The median RT usefulness could be established for both test conditions (FAST_MOT and FAST_SAT) from the third appointment at the latest. These findings highlight the necessity of incorporating adequate familiarization trials before testing to ensure useful results. The RAI showed only poor reliability and, as the TE exceeded the SWC 0.5, the usefulness cannot be assumed for this evaluated parameter. Calculated indices were previously used to describe the differences in performance between preplanned and reactive COD runs and, therefore, to estimate speed and perceptual abilities of athletes [32,43,49,50]. These ratios were suggested to use for recommendations regarding training content [32]. The results of this study propose being careful when relying only on the results of the ratio of two tests to prescribe training recommendations due to the low reliability and low sensitivity for performance change (usefulness). Following Gabett and colleagues (2008), a further calculation of the ratio of pre-planned and reactive COD runs could also be computed to categorize athletes in relation to the average team results [51]. The resulting ratio allows classification into four categories: “slow thinker/fast thinker” and “slow mover/fast thinker” [51]. This classification could also prescribe specific training content with different foci related to agility’s motor and cognitive subcomponents. However, interventional studies are needed to test the efficiency of such approaches.

### 4.4. Median Analysis in Agility Research

Apart from the analysis of validity, reliability, and usefulness of the FAST, the present study demonstrated that the results between mean and median values can lead to different estimates of reliability and usefulness. In our study, mean RTs of the FAST showed only maximal moderate reliability, and usefulness could only be proven for the FAST_MOT from the third session onwards. In contrast, for the median RTs, moderate to good reliability could be attested and usefulness could be established from the first session for the FAST_MOT and from the third session for the FAST_SAT. Therefore, reporting both mean and median could lead to a better understanding of the data and the distribution of RTs in agility tests [52]. However, in most published studies for agility research, mean values are reported to represent the central tendency. However, outliers impact mean values more and can be distorted in non-normally distributed data [53,54]. Even when values resulting from technical errors have been excluded, the RTs for each stimulus per condition of all participants were not normally distributed and showed a right-skewed distribution. Although the mean is the most used measure of central tendency, individual RTs are often not examined in detail. Instead, the mean value of individual responses per trial is calculated, and then a group mean is derived from these trial means. Since individual stimulus responses are likely to be skewed, reporting median values rather than mean values [53,54,55] would be advisable. Therefore, future studies should consider analyzing RTs in more detail and then decide which metric of the central tendency is the most suitable, or report mean and median RTs for a more transparent inside in the data distribution.

### 4.5. Limitations and Directions for Further Research

The results of the study should be interpreted with several limitations in mind. Firstly, since validation was conducted exclusively with female athletes, future studies should include male athletes to account for gender-specific differences in the FAST [32]. Additionally, expertise [56,57,58] and sport disciplines [22,59] have been shown to influence agility performance. Therefore, future studies may incorporate larger and more diverse samples to compare sport-specific differences in FAST performance. Although RTs have already been selected as parameters in this study, which provide a more detailed statement than total times [12], this is the only parameter used to describe agility performance. Further analyses of reaction times (time from stimuli presentation to movement initialization), decision accuracy, acceleration and deceleration velocities [12], or visual screening behavior [44] could be used to make detailed statements about the behavior. Secondly, all measurements occurred indoors on the surface of the affiliated university’s SpeedCourt system (GlobalSpeed GmbH, Hembach, Germany). This synthetic sports hall surface only mirrors the natural sports surface of some included indoor sports athletes like handball or basketball athletes. For other participants who perform outdoor sports, like soccer players, the surface was less familiar. It could have led to adapted running behavior due to unknown ground conditions or missing experiences on the ground when changing direction quickly [56,60].

As the Fitlight system allows custom configurations, the complexity of the requirements for the athletes can be increased to advance further approximations to the sporting environment. Different stimuli can be programmed for processing, allowing the testing of simple reactions and choice reactions, inhibition, and further complex tasks. For instance, color-coded running patterns can be implemented, where different colors correspond to specific instructions. This ensures that the signaling light does not always indicate the target light to be approached and, therefore, athletes would need to inhibit the initial response of directly approach the stimuli light. This would enable the mapping of athletes’ decision-making behavior in more complex situations [14]. To enhance ecological validity further, future tests could be conducted outside the laboratory in the athletes’ actual sports environment, which is possible due to the portability of the measurement equipment. This approach would encourage more natural running behavior, as the athletes would perform the test on the same surface where they typically train and compete.

## 5. Conclusions

In the present study, we showed that the newly developed FAST can discriminate between GS and NGS athletes and is, therefore, a valid test battery for game sport-specific agility. In addition, we were able to show that the level of reliability varies depending on the consideration of mean or median RT values, which results in different classifications of test–retest reliability (moderate to good) and varying statements about usefulness. As a result, it should be considered in future whether the use of mean and median values should be used to describe performance.

## Figures and Tables

**Figure 1 jfmk-10-00126-f001:**
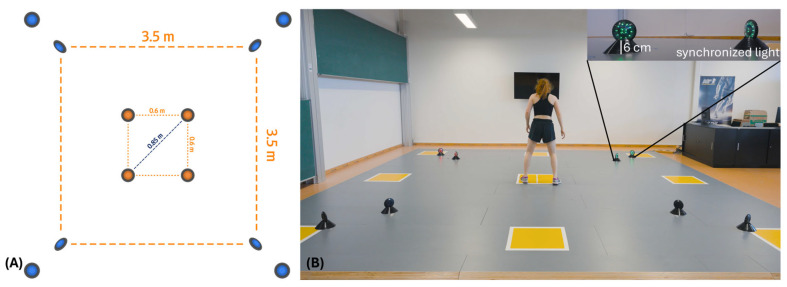
Schematic representation of the Functional Agility Square Test (FAST). (**A**) Smaller square (0.6 m × 0.6 m, diagonal of 0.85 m) shows the setup of the FAST cognitive (FAST_COG) and starting position for FAST motor (FAST_MOT) and FAST “Simple Agility Test” (FAST_SAT). Outer square (3.5 m × 3.5 m, diagonal of 5.0 m) shows the setup of the FAST_MOT and FAST_SAT. (**B**) Realistic demonstration of FAST_MOT and FAST_SAT setup with the participant standing in the middle position. In each corner, two lights were placed, with the front one facing the side to step and the rear one facing the center position. Each condition was performed three times.

**Figure 2 jfmk-10-00126-f002:**
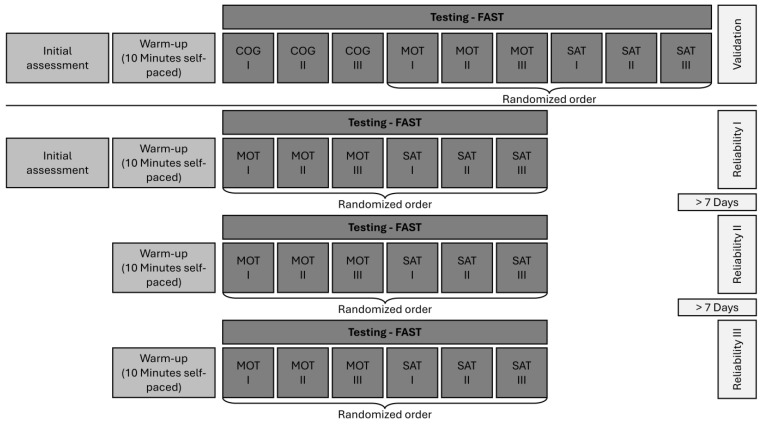
Overview of the experimental protocol. For validation purpose participants performed the cognitive (FAST_COG), motor (FAST_MOT), and “Simple Agility Test” (FAST_SAT) three times in one session. For reliability testing, participants came in on three days with at least seven days’ break between the sessions. Reliability sessions consisted of three trials of the FAST_MOT and the FAST_SAT in a randomized order.

**Table 1 jfmk-10-00126-t001:** Overview of demographic information of the sample of game sports (*n* = 22) and non-game sports (*n* = 22) athletes included in discriminant validity analysis in the first three columns. For reliability, demographic information of the sample of athletes (*n* = 36) was included in test–retest reliability analysis, and is displayed in the last column.

	Validation			Reliability
	Game Sports	Non-Game Sports		Game Sports
	Mean ± SD		*p*	Mean ± SD
Female/Male	22/0	22/0		20/16
Age (years)	23.4 ± 3.0	23.7 ± 3.4	0.709	23.39 ± 2.93
Height (cm)	169.4 ± 5.3	166.3 ± 6.8	0.098	175.36 ± 8.41
Weight (kg)	63.2 ± 8.7	59.8 ± 11.2	0.267	69.09 ± 10.10
Sporting experience (years)	15.8 ± 3.5	11.8 ± 5.7	0.009 *	14.23 ± 5.51
Sports	Soccer (18) Handball (3) Ultimate Frisbee (1)	Track and Field (6) Dance (3) Fitness (3) Gymnastics (3) Swimming (2) CrossFit (1) Equestrian sports (1) Kickboxing (1) Rhythmic gymnastics (1)		Soccer (13/14) Handball (4/0) Basketball (0/1) American Football (0/1) Ultimate Frisbee (1/0) Volleyball (0/1) Tennis (0/1)

* Significantly different (*p* < 0.05).

**Table 2 jfmk-10-00126-t002:** Comparison of mean and median motor response times between game sports (GS; *n* = 22) and non-game sports athletes (NGS; *n* = 22) in the cognitive (COG), motor (MOT), simple agility test (SAT) of the Functional Agility Square Test (FAST), and the reactive agility index (RAI).

		MEAN					MEDIAN				
		Mean	SD	*p*	t/U	ES	Mean	SD	*p*	t	ES
FAST_COG	GS	523.15	45.44	0.023 ^MWU^ *	145 ^MWU^	−0.34	529	42.77	0.009 *	−2.47	−0.74
	NGS	575.78	78.09				573.55	73.02			
FAST_MOT	GS	1097.37	89.79	0.013 *	−2.32	−0.7	1094.45	90.67	0.018 *	−2.17	−0.65
	NGS	1179.11	139.06				1173.36	144.78			
FAST_SAT	GS	1458.71	101.62	0.010 *	−2.40	−0.73	1410.82	82.62	0.023 *	−2.06	−0.62
	NGS	1546.23	137.21				1480.05	134.17			
RAI	GS	361.34	110.86	0.438	−0.157	−0.047	316.36	99.17	0.382	−1.46	0.091
	NGS	367.12	131.79				306.68	112.13			

Abbreviations. FAST_COG: Functional Agility Square Test Cognitive; FAST_MOT: Functional Agility Square Test Motor; FAST_SAT: Functional Agility Square Test Simple Agility Test; GS: game sports; ^MWU^: Mann–Whitney U Test; NGS: non-game sports; RAI: Reactive Agility Index; * significantly different (*p* < 0.05).

**Table 3 jfmk-10-00126-t003:** Test–retest reliability for mean response times in milliseconds for motor (FAST_MOT), simple agility test (FAST_SAT) and reactive agility index (RAI) of three sessions of game sports athletes (*n* = 36).

	MEAN									
	Session I RT	Session II RT	ICC [LB UB]	SEM [LB UB]	CoV (%) [LB UB]	Session II RT	Session III RT	ICC [LB UB]	SEM [LB UB]	CoV (%) [LB UB]
FAST_MOT	1088.69	1058.70	0.60 [0.35–0.77]	78.17 [99.95–58.90]	7.28 [9.31–5.49]	1058.70	1032.85	0.74 [0.54–0.86]	59.36 [78.48–43.77]	5.68 [7.50–4.19]
FAST_SAT	1427.67	1389.99	0.57 [0.29–0.75]	68.91 [88.05–51.94]	4.89 [6.25–3.69]	1389.99	1369.75	0.50 [0.22–0.71]	76.63 [96.18–58.49]	5.55 [6.97–4.24]
RAI	338.98	331.29	0.56 [0.28–0.75]	85.03 [108.35–64.22]	25.37 [32.33–19.16]	331.29	336.90	0.44 [0.13–0.67]	84.10 [104.77–64.56]	25.17 [31.36–19.32]

Abbreviations. CoV: Coefficient of Variation; FAST_MOT: Functional Agility Square Test “Motor”; FAST_SAT: Functional Agility Square Test “Simple Agility Test”; ICC: Intraclass Correlation Coefficient; LB: Lower Bound; RAI: Reactive Agility Index; RT: Response Time; SEM: standard error of measurement; UB: Upper Bound.

**Table 4 jfmk-10-00126-t004:** Test–retest reliability for median response times for motor (FAST_MOT), simple agility test (FAST_SAT) and reactive agility index (RAI) of three sessions of game sports athletes (*n* = 36).

MEDIAN										
	Session I RT	Session II RT	ICC [LB UB]	SEM [LB UB]	CoV (%) [LB UB]	Session II RT	Session III RT	ICC [LB UB]	SEM [LB UB]	CoV (%) [LB UB]
FAST_MOT	1058	1059	0.71 [0.51–0.84]	60.47 [79.22–44.84]	5.61 [7.35–4.16]	1059	1033	0.83 [0.67–0.91]	47.20 [65.38–33.88]	4.48 [6.21–3.22]
FAST_SAT	1362	1358	0.77 [0.59–0.87]	43.65 [57.79–32.16]	3.20 [4.24–2.36]	1358	1356	0.59 [0.33–0.76]	61.12 [77.87–46.16]	4.54 [5.78–3.43]
RAI	287	275	0.57 [0.30–0.75]	68.06 [86.72–51.40]	23.87 [30.41–18.02]	275	321	0.44 [0.14–0.67]	70.20 [87.56–58.83]	23.98 [29.91–18.39]

Abbreviations. CoV: Coefficient of Variation; FAST_MOT: Functional Agility Square Test “Motor”; FAST_SAT: Functional Agility Square Test “Simple Agility Test”; ICC: Intraclass Correlation Coefficient; LB: Lower Bound; RAI: Reactive Agility Index; RT: Response Time; SEM: standard error of measurement; UB: Upper Bound.

**Table 5 jfmk-10-00126-t005:** Measures of usefulness of the motor (FAST_MOT) and simple agility test (FAST_SAT) of the three sessions for mean and median response times.

		MEAN			MEDIAN		
	Parameter	Session I	Session II	Session III	Session I	Session II	Session III
FAST_MOT	TE SWC 0.5 Rating	134.7 104.3 Marginal	89.9 73.3 Marginal	55.1 61.3 Good	56.8 63.9 Good	46.6 57.5 Good	40.4 58.9 Good
FAST_SAT	TE SWC 0.5 Rating	122.3 76.4 Marginal	104.1 67.5 Marginal	78.7 70.9 Marginal	82.2 55.8 Marginal	78.7 59.3 Marginal	44.4 54.5 Good

Abbreviations. FAST_MOT: Function Agility Square Test “Motor”; FAST_SAT: Functional Agility Square Test “Simple Agility Test”; SWC 0.5: Smallest Worthwhile Change 0.5; TE: Typical Error.

## Data Availability

The data that support the findings of this study are available from the corresponding author upon reasonable request.

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
