# Peer review of "Validation, Reliability, and Usefulness of the Functional Agility Square Test [FAST]"

_jfmk, 2025, doi:10.3390/jfmk10020126_

Round 1
Reviewer 1 Report
Comments and Suggestions for Authors
The manuscript entitled "Validation, Reliability, and Usefulness of the Functional Agility Square Test [FAST]" demonstrates a new test to assess agility in Sport Science. This is an excellently designed and well-written study. The introduction is concise but exhaustive. The methods are described very well, which allows for replication of the study. The results are clearly shown in tables and the text. Also, the discussion is well-written. Overall, this is an excellent article describing an essential and scientifically valid matter. Thus, I congratulate the authors on it.
I have only two suggestions to improve that manuscript:
- Please report the test statistic (t or U, respectively) effect size for Student's t-test and Mann-Whitney U-test (ideally in Table 2). Please also add the interpretation of the effect size in the text.
- Please add to the limitations section that the study was conducted on a sample of women and that it is necessary to verify the same studies in a group of men in the future and compare the results between the sexes. Similarly, age and sports experience can be crucial for performance on the FAST. It would also be good to standardize the test in the future and check whether the same norms will apply to women and men, as well as athletes with different sports experience and at different ages (children, adolescents, adults). In addition, various sports disciplines can be crucial for performance as well. Therefore, comparing football with other team disciplines as separate groups is recommended in the future.
Reviewer 2 Report
Comments and Suggestions for Authors
Thank you for the opportunity to evaluate this scientific article. In an effort to synthesize, my recommendations for correcting the informational content for each chapter are the following:
Introduction
It would be appropriate to make some broader references about the limitations of current agility tests and the need for a new test.
Methods
I consider it necessary to detail the selection criteria for game athletes (GS) and non-sports athletes (NGS), in order to clarify the relevance of the sample.
In the section on procedures, a rationale could be added for each method used, explaining why it is suitable for measuring agility, minimal explanations.
Results
Instead of presenting only statistical values, it would be appropriate to explain in simple terms what these results mean for athletic performance.
Discussion
Although it is mentioned that future studies could increase the complexity of the tasks, it would be useful to suggest concrete directions for improvement.
Conclusion
no comments
